# Cancer-Related Alopecia: From Etiologies to Global Management

**DOI:** 10.3390/cancers13215556

**Published:** 2021-11-05

**Authors:** Stanislas Quesada, Alexandre Guichard, Frédéric Fiteni

**Affiliations:** 1Institut Régional du Cancer de Montpellier, 34090 Montpellier, France; 2Medical Oncology Department, University Hospital, 30900 Nîmes, France; Frederic.FITENI@chu-nimes.fr; 3Faculty of Medicine, University of Montpellier, 34090 Montpellier, France; 4Legacy Healthcare Switzerland SA, 1066 Epalinges, Switzerland; guichard.alexandre@gmail.com; 5UMR INSERM IDESP—Desbrest Institute of Epidemiology and Public Health, University of Montpellier, 34090 Montpellier, France

**Keywords:** alopecia, supportive care, psycho-oncology, quality of life

## Abstract

**Simple Summary:**

Although it does not represent a condition that threatens the life of patients, alopecia nevertheless has an essential impact on the quality of life of patients, particularly in terms of the psychological and social aspects. Indeed, while it has long been considered an acceptable side effect in the management of patients, the progressive emergence of a patient-centered approach coupled with a better knowledge of the pathophysiological processes involved has led to a better consideration of alopecia, both on the preventive and palliative sides. Thus, cancerous alopecia can be multifactorial: iatrogenic (in particular via conventional chemotherapy), induced by a vitamin/nutritional deficiency, or even caused by the disease itself. In this state-of-the-art review, we therefore cover alopecia in an exhaustive manner by considering the different mechanisms involved and their frequency as well as the various therapies offered.

**Abstract:**

Alopecia represents a multifaceted challenge with distinct etiologies and consequences. Transposed to the world of oncology, different types of alopecia and molecular pathways have been characterized, allowing a better understanding of the underlying mechanisms. In patients with cancer, alopecia can be iatrogenic (i.e., due to conventional chemotherapies, endocrine therapies, targeted therapies, immunotherapies, radiotherapy and surgery) or a direct consequence of the disease itself (e.g., malnutrition, scalp metastases and paraneoplastic syndromes). Identification of the actual incriminated mechanism(s) is therefore essential in order to deliver appropriate supportive care, whether preventive or curative. On the preventive side, the last few years have seen the advent of the automated cooling cap, a prophylactic approach supported by several randomized clinical trials. On the curative side, although the treatments currently available are limited, several promising therapeutic approaches are under development. Appropriate alopecia management is essential, particularly regarding its psychological repercussions with significant consequences on the quality of life of patients and their family and with a potential impact on treatment compliance.

## 1. Introduction

Alopecia, which is defined as a decrease in hair density, exhibits a wide range of features. Indeed, it may be localized, diffuse or total, acute or chronic, sudden or gradual, reversible or permanent. Alopecia might be considered as a common symptom to several pathologies with varied etiologies (e.g., immunological, inflammatory and infectious). In addition, alopecia can be the sentinel sign of systemic diseases: endocrine (e.g., thyroid dysfunction), autoimmune (e.g., systemic lupus erythematosus), psychiatric (e.g., trichotillomania) or infectious. Alopecia can be classified into three classes: non-scarring (such as androgenetic alopecia—AGA, which is the most common alopecia), scarring (with destruction of the hair follicle) and congenital alopecia [1].

Understanding hair alterations requires knowledge of its physiology. Briefly, hair is a complex mini-organism with rapid renewal, a dense vascular network and immune privilege. The hair’s life cycle is schematically divided into four phases: anagen (4–6 years growth), catagen (3 weeks with massive apoptosis), telogen (3 month resting followed by expulsion) and kenogen (2–12 months latency). Hair follicles are asynchronous in humans, with ≈85% of anagen hair, ≈1% of catagen hair and ≈15% of telogen hair. Hair loss is considered as pathological when it represents more than 150 hairs per day. Importantly, hair loss is not necessarily accompanied by a visible decrease in hair density. Of note, effluvium corresponds to a sudden, abundant and diffuse hair loss, which can be acute or chronic and affect anagen or telogen hair [1]. 

Although it is not life threatening, alopecia represents one of the most important parameters affecting the quality of life (QoL) of patients with cancers, particularly in terms of the psychological and social aspects [2,3,4]. Notably, caregivers tend to underestimate the impact of alopecia on patients [5]. Indeed, while it has long been considered an acceptable side effect in the management of patients, the increasing number of cancer survivors coupled with a better knowledge of the pathophysiological processes involved has led to a better consideration of alopecia, both on the preventive and palliative sides.

To our knowledge, although there are several excellent reviews focused on a specific aspect of cancer-related alopecia (e.g., reviews assessing mechanisms involved in hair disorders in cancer, notably through anticancer therapies), there is not currently an integrative review that could help physicians and healthcare providers involved in the supportive care in cancers in gaining a global approach.

As such, the scope of this review is to provide physicians with a state-of-the-art and clinical practice-driven review, assessing in a comprehensive manner the distinct mechanisms involved in cancer-related alopecia, their respective frequencies and the current and future treatment approaches.

## 2. Cancer-Related Alopecia: Mechanisms and Epidemiology

### 2.1. Classification and Diagnosis

In spite of being a common side effect of many cancer therapies, its clinical presentations, frequencies and underlying mechanisms are plural and mainly related to the type of treatment used, without distinction of gender or age. Clinically, alopecia can be accompanied by dysesthesia, pruritus and dryness of the scalp [6]. It is not restricted to the scalp and can therefore affect other body hair, such as the eyebrows, eyelashes and axillo-pubic hair. Table 1 shows the main characteristics of these attacks according to the treatment used [7,8,9,10,11,12,13,14].

Schematically, it is possible to classify antineoplastic-induced alopecia (ANIA) into three groups (which can be overlapping) according to their mechanism: follicle destruction (mainly chemotherapy and radiotherapy), follicle miniaturization (mainly endocrine and targeted therapies) and hair cycle blockage (mainly immunotherapy). By knowing the exact mechanism of action, the clinician is able to adjust the management of alopecia. 

Alopecia diagnosis is based on both anamnesis and clinical presentation, which becomes visible when hair density loss is greater than 50%. ANIA and other hair disorders share common clinical patterns. A macroscopic examination with dermoscopy (called “trichoscopy” when applied to the scalp) represents an important tool of diagnosis and prognosis, allowing a finer assessment of hair density, thickness and hair shaft anomalies. In addition, the trichogram, a semi-invasive quantitative technique consisting of a microscopic analysis of a plucked hair’s bulb, allows the quantification of anagen and telogen hairs, which may be relevant for ANIA diagnosis and to perform differential diagnosis [15].

The severity of ANIA is mainly scored with the Common Terminology Criteria for Adverse Events (CTCAE) classification and the Severity of Alopecia Tool (SALT) score (Table 2) [16,17]. The SALT score, defined by Olsen and colleagues, is expressed as a percentage of hair loss and can be used either as a continuous or as a categorical variable (Figure 1).

CTCAE in its version 5.0 defines two grades of severity. Grade 1 refers to hair loss of <50%, which is not obvious from a distance but only on close inspection, while grade 2 ties in with hair loss of ≥50% normal for that individual, which is readily apparent to others. While grade 1 can be hidden with a different hairstyle, grade 2 alopecia requires a wig if the patient desires to completely camouflage the disorder; furthermore, grade 2 alopecia is likely to lead to an impact on QoL. As such, QoL needs to be evaluated, notably in the context of severe and/or persistent alopecia. Two auto-questionnaires assessing the psychological distress induced by alopecia have been developed: the *Chemotherapy-Induced Alopecia Distress Scale* (directly assessing PDIA) and the *Hairdex* assessing alopecia-related QoL [2,18]. Importantly, the *EORTC QLQ-BR45* includes an item on alopecia (item 34: “Have you been bothered by hair loss?”) [19].

### 2.2. Chemotherapy

In spite of a frequency of ≈65% (all chemotherapy protocols combined) and the significant impact on patients’ lives, research on chemotherapy-induced alopecia (CIA) has remained elusive until recently [20]. CIA appears sub-acutely quickly after the chemotherapy protocol initiation and turns maximal within a few weeks. After chemotherapy, hair growth will resume a normal rhythm within 3 months and normally reach an aesthetically suitable result at 6 months. Predominance in frontal and occipital regions is classically observed [16]. Molecularly, several pathways have been characterized, eventually leading to a massive apoptosis [21]. The mechanism is mainly an anagen effluvium, although telogen effluvium may also be observed [22].

The main predictive factor of CIA is the chemotherapy class (Table 3): alkylating agents, anthracyclines, taxanes and etoposide are the ones with the most common and severe effects [10,11]. Conversely, some molecules infrequently cause CIA; notably, molecules such as fluorouracil and methotrexate cause more patchy alopecia. Beyond the therapeutic class, various predictive factors are known: polychemotherapy regimen and dosage, concomitant anticancer treatment, nutritional and hormonal statuses and the presence of an underlying AGA [23]. Furthermore, a CIA risk stratification based on genomics is emerging, which could pave the way for better identification of high-risk patients in order to provide them with appropriate supportive care [24].

### 2.3. Focus on Persistent CIA (pCIA)

Although classically considered as a temporary mechanism, several studies have shown that CIA might be persistent [10,20,25]. Initially described in hematology through allograft conditioning before bone marrow transplant [26], pCIA was later reported with solid tumors [27]. This latter discovery is manifested by a lesser knowledge of pCIA by oncologists compared to dermatologists [28]. pCIA is defined by the presence of alopecia beyond 6 months after chemotherapy completion. It can exhibit various clinical aspects: mostly diffuse and non-scarring (≈50% of cases), with possible scarring involvement.

Histologically, destruction of the follicular epithelial stem cell pool and follicular miniaturization are the main suspected mechanisms [25]. Actually, this is not an uncommon phenomenon, especially in patients treated for breast cancer (BC) with taxane-based protocols [11]. Indeed, a prospective study has shown that BC patients treated with taxanes exhibited pCIA in ≈40% at 6 months, with persistence at 3 years [29]. These recent data reinforce previous observations, both retrospectively and prospectively [11].

Furthermore, pCIA also exhibits modifications in hair quality: indeed, it has been estimated that up to 75% of patients with pCIA still had hair thinning at 3 years post-chemotherapy [30]. Notably, an association has recently been shown between a regulatory portion of the *ABCB1* gene and pCIA in patients with BC treated with taxane-based chemotherapy [31]. Specific attention should be given after bone marrow transplant pCIA in pediatric oncology, where a frequency of ≈20% has been reported [32].

### 2.4. Targeted Therapies (TTs)

TTs cover very different mechanisms of action and targets and their impact on alopecia mainly depends on the molecular target(s). TT-induced alopecia (TIA) confirms the necessary activation of several signaling pathways, such as *SHH*, *EGFR* and *VEGF*, in hair physiology [1].

Way different from CIA, TIAs exhibit specific profiles and evolutions [12]. Indeed, TIA does not appear suddenly and can regress through treatment course, with a whole range of hair modifications: texture, density, color and renewal rate [10,13]. According to a 2015 meta-analysis, TIA affects ≈15% of patients, with a molecule-dependent high variability (Table 4) [14]. 

Some molecules need to be outlined. Importantly, vismodegib (an SHH inhibitor) has the highest TIA rate, with an estimated frequency of ≈60%; furthermore, cases of persistent TIA have been reported [33,34]. Anti-EGFR molecules present a certain risk of non-scarring alopecia, sometimes completed with a scarring pattern as a consequence of the iatrogenic anti-EGFR facial papulo-pustulosis. Furthermore, it should be noted that the hair has a dry, brittle and curly appearance; in addition, patients show characteristic hypertrichosis and trichomegaly [13]. BRAF inhibitors lead to initial alopecia in ≈20% of patients, although hair regrowth may occur despite continued treatment [35]. Notably, while the use of trametinib (MEK inhibitor) leads to alopecia in about 13% of cases, dual BRAF/MEK inhibition with vemurafenib/cobimetinib or dabrafenib/trametinib leads to alopecia in 13% and 6% of cases, respectively [13,14].

Interestingly, a phase 2 study evaluating bevacizumab monotherapy in patients with angiosarcoma reported a ≈10% incidence of alopecia, although this molecule is classically considered as not causative of alopecia [36]. With the “multi-targeted” protein kinase inhibitors sorafenib and regorafenib, the risk of alopecia should be considered, since frequencies of 25–30% have been reported [14,37]. Concerning emerging PARP inhibitors, no specific TIA has been reported to date [10].

### 2.5. Endocrine Therapies (ETs)

ET-induced alopecia (EIA) has mainly been described in the context of hormone receptor-positive BC; indeed, the high prevalence of BC and the ET prescription length led to a fine characterization of this toxicity. Molecularly and clinically, EIA emerges in a different way, since it appears progressively and exhibits an AGA-like pattern [8]. The EIA overall incidence—assessed via a large meta-analysis—is ≈5%, with significant variation depending on the therapeutic class(es) used (Table 5) [38]. Notably, total alopecia has been described in patients treated with tamoxifen [39]. Importantly, EIA can be a vector of non-compliance; indeed, it has been reported that 8% of patients stopped taking aromatase inhibitors because of EIA [40]. Notably, CDK4/6 inhibitors (used concomitantly with ET in BC) seem to potentiate EIA [41,42]. In the context of prostate cancer, none of the androgen deprivation therapies have been associated with EIA to date [9].

### 2.6. Radiotherapy (RT)

Radiation-induced alopecia (RIA) has to be considered in two situations: central nervous system primary tumors and brain metastases. Apart from stereotactic RT, the classical treatment of brain metastases is the “pan-encephalic” RT (PERT) protocol. Historical studies have determined threshold doses per fraction: 0.75–2 Gy for temporary depilation and 8–16 Gy for hair follicle sterilization. Mechanistically, RIA consists of an anagen effluvium. Numerous predictive factors of RIA have been characterized: doses (per fraction and total), the type of ionizing radiations (photons vs. protons), the surface and volume of irradiation, concomitant treatment, hair capital and the genetic constitution of the patient [7]. 

RIA appearance is relatively abrupt and occurs within 1–3 weeks after treatment initiation. It concerns 75–100% of PERT-treated patients (since the dose per fraction is >2 Gy) with a regrowth around 2–4 months post-protocol [43].

Persistent RIA (pRIA) is defined as the presence of alopecia over 6 months post-RT; it is estimated to occur in 60% of PERT-treated patients, notably through scarring alopecia. Different predictive irradiation thresholds have been proposed: from ≈21 Gy in children (treated with concomitant high-dose chemotherapy) to ≈43 Gy in adults [44,45]. Recently, a study suggested a 36 Gy threshold [7]. The evolution of RT protocols and new technologies will certainly lead to a revision of radiotoxicity data in the future.

### 2.7. Immunotherapy

Immune checkpoints function by maintaining immunological homeostasis through the inhibition of T-cell activation. Immune checkpoint inhibitor (ICI) actions lead to constitutive T-cell activation and anti-tumor activity; however, they are counterbalanced by a range of dysimmune toxicities grouped as immune-related adverse events (IRAEs) that can affect virtually any organ [13,46]. Skin toxicities, including maculopapular rash, eczema and vitiligo, are the most common IRAEs, affecting ≈40% of patients [46]. Notably, some of these skin conditions may lead to alopecia when they extend to the scalp.

Currently, ICPI-induced alopecia (IIA) is estimated at 1–2%, with a molecule-dependent variation [47]. In a recent meta-analysis focused on melanoma treatment, the following incidences of alopecia were found: 1.7% for ipilimumab, 2% for nivolumab and 3.4% for pembrolizumab [48]. According to the mechanism involved, namely direct or indirect, IIA can be classified as primary (IIA-P) or secondary (IIA-S), respectively.

IIA-P occurs when the reactivation of ICPIs directly leads to scalp dysimmunity. Mechanistically, IIA-P exhibits *alopecia areata* features; the hair follicle loses its immune privilege, with an intense perifollicular lymphocytic infiltration. Concerning onset kinetics, high variability has been described, from a few weeks to over a year [11]. Persistent IIA has already been described through case reports [49]. To date, treatment mainly involves class IV topical steroids or systemic immunosuppressants in recalcitrant cases [47]. Very interestingly, case reports proposed that IIA-P could be a predictive marker of nivolumab efficacy in melanoma [50]. Beyond IIA, changes in texture and hair repigmentation processes have also been described [46].

IIA-S should be observed in the context of IRAE, as one of its indirect consequences is alopecia. Therefore, it is necessary to rule out an underlying IIA-S before diagnosing an IIA-P. IIA-S management has to consider an etiologic therapy and to favor a multidisciplinary approach [51]. The most often identified IIA-S is through thyroid dysfunction; indeed, it is estimated that ≈10% of ICPI-treated patients will develop thyroid complications [46]. Emerging upon ICPI introduction, genuine autoimmune (e.g., systemic lupus erythematosus and scleroderma) and inflammatory (e.g., sarcoidosis-like and severe drug conditions) diseases have been reported, leading to secondary scarring alopecia [52]. 

### 2.8. Other Mechanisms

A certain number of mechanisms potently leading to alopecia are worth mentioning. The first one is cancer-induced malnutrition, affecting 30–50% of patients [53]. Moreover, hair renewal requires a sufficient vitamin/mineral/energy supply; for instance, vitamin (e.g., vitamin D) and/or micronutrient (e.g., iron and zinc) deficiencies can lead to alopecia [54]. Apart from these, alopecia may unravel an underlying cancer by two main mechanisms. Firstly, scalp metastases can exhibit alopecia features, the so-called *alopecia neoplastica* [55]. Secondly, a paraneoplastic mechanism can be the first clinical sign of cancer; several case reports of paraneoplastic alopecia have been described, notably through paraneoplastic dermatosis [56,57]. As previously described, alopecia has been described following allogeneic BMT via graft-versus-host disease [58].

## 3. Alopecia Management 

To date, no curative treatment is indicated for ANIA; however, several studies have demonstrated some effectiveness of certain treatments. Global alopecia management (from prophylaxis to palliative approaches) is summarized in Table 6 [10,11,12,13,59].

Importantly, although it is clearly out of the scope of this review, psychological alterations related to alopecia and their consequences on quality of life should not be overlooked [7,20,25].

### 3.1. ANIA Prevention

As the number of patients cured or in remission is constantly growing and with the move toward a patient-centered medicine, supportive care concerning ANIA appears essential [25,59]. 

Initial management, before any anticancer treatment, is hair status evaluation. Indeed, a pre-existing pathological condition (such as vitamin/mineral deficiency or a more general disorder) is likely to increase ANIA, either by a complementary mechanism or in synergy. Although there is no standard test currently recommended, the following minimum biological test should be performed when facing alopecia without an obvious etiology: complete blood count, [TSH], [vitamin D], [iron] +/− hormonal assays.

In the therapeutic arsenal, the cooling cap (CC)—with or without cooling mittens/socks in order to protect extremities—has been used empirically for several decades. CC is based on cooling the scalp during chemotherapy administration in order to provoke a local vasoconstriction and, thus, a lesser exposure of the hair follicles. Two techniques currently exist: CC filled with gel and kept cold (gCC, requiring a regular change of CC during the same cure) and electric CC (eCC, based on the circulation of a capillary cooling liquid through an automated, meanwhile more expensive, technique). CC was quickly considered relevant because of its non-invasiveness and its low cost, counterbalanced by a formal absence of proof of effectiveness and safety. Nevertheless, democratization of its use has been slowing down for a long time, considering the theoretical risk of scalp metastases, the multiplicity of care practices, protocols with a long IV infusion and patient acceptability [60]. Indeed, historical reports of a few cases of scalp metastases have been reported, leading to an initial precautionary principle. Since the 1980s, numerous exploratory studies have been performed, primarily in patients with BC; however, the lack of randomized clinical trials and the multiplicity of chemotherapy protocols and of ways to use CC led to limited data [61]. 

Meanwhile, two prospective clinical trials published in 2017 have shown the efficacy of eCC. The first one was a prospective multicenter cohort study of patients (106 using the DigniCap© eCC and 16 as controls) with localized BC and treated with adjuvant taxane-based chemotherapy and followed-up annually for 5 years. The primary outcome was whether the use of eCC was associated with ANIA prevention (defined as CTCAE grade ≤1 alopecia, equivalent to hair loss ≤50%): indeed, two-thirds of the patients in the eCC group had ANIA prevention versus 0% in the control group. Furthermore, the impact on QoL was also assessed: only one-quarter of patients felt “physically less attractive” in the eCC group versus >50% in the control group [62]. The second study evaluated the efficacy of the PaxMan© eCC on CIA in a multicenter randomized clinical trial, which enrolled 119 patients in the eCC group (versus 63 in the control group) with localized BC and treated with anthracyclines and/or taxanes. The primary endpoint was ANIA prevention (defined as CTCAE grade ≤1 alopecia) after four cycles of chemotherapy. Hair preservation was found in 50% (CI95% = 40.7–60.4%) of patients in the eCC group versus 0% (CI95% = 0–7.6%) in the control group [63]. In addition, a meta-analysis (pooling 24 studies concerning BC patients) did not find an increased risk of scalp metastases with CC [64]. 

Furthermore, meta-analyses have shown that eCC exhibits more efficacy compared to other techniques available for ANIA prevention [65,66]. Importantly, CC is currently the only treatment validated by the FDA for CIA prophylaxis and could therefore represent an effective and safe tool for some chemotherapy protocols. However, due to the current dynamics of the ambulatory shift, chemotherapies given through a long IV infusion as well as those dispensed orally are not eligible for the use of CC [67].

### 3.2. ANIA Treatments

Topical minoxidil (TMX) is indicated in AGA in both genders. Recently, TMX-5% has been demonstrated to be efficient in EIA (*n* = 46), with 80% of patients showing moderate to significant improvement within 3–6 months [8]. Furthermore, TMX-2% has shown some efficacy after chemotherapy for hair regrowth (by reducing CIA from 137 days for placebo to 87 days), without an effect on CIA prevention [68]. In a cohort of patients with pRIA (*n* = 34), TMX-5% exhibited some effect, with 12% complete response and 38% partial response rates [7].

Spironolactone (owing to its anti-androgenic effect) tends to be prescribed off-label for female AGA when accompanied by signs of hyperandrogenism. Since AGA and EIA share the same pathophysiology, spironolactone could represent an interesting treatment of EIA. Concerning its safety, a recent study has shown that spironolactone did not interact with ET and did not increase the risk of BC [69].

Bimatoprost (a synthetic prostaglandin analogue) is classically indicated as an eye drop for intraocular hypertonia. Since 2008, it is also indicated for ciliary hypotrichosis treatment. Several controlled clinical studies have shown bimatoprost gel efficacy for chemotherapy-induced ciliary hypotrichosis, reporting a faster regrowth and an increased density of treated lashes [70].

Hair autotransplantation (HAT), a surgery technique, consists of harvesting follicles from a donor area and then transplanting them into a recipient area in order to increase hair density; in the context of ANIA, the main limit is an insufficient follicular density of the donor area. Recently, it has been demonstrated that HAT could be a relevant treatment for EIA in women: interestingly, a single session was sufficient to recover a satisfying hairline in 70% of patients. In addition, a 3-year follow-up confirmed the persistence of the graft as well as its safety [71]. This technique could be extended to other ANIAs when the donor area is sufficient and when alopecia is considered stable. Finally, a skin expansion (by placing prostheses under the scalp and followed by plastic surgery) could be a promising surgical technique for pRIA/pCIA [72].

Regarding the therapeutic perspectives, various preclinical and clinical studies have reported the efficacy of molecules such as cyclosporine, topical vasoconstrictors and antioxidants [73]. However, large-scale transposition in humans has not yet demonstrated clinically significant efficacy. The Clinicaltrials.gov database currently references 15 active clinical studies for ANIA treatment [74]. For CIA, studies include CC (*n* = 10), keratinocyte growth factor (*n* = 1) and LED lights (*n* = 1). One study is currently evaluating the effect of platelet-rich plasma in EIA and pCIA. For pCIA, ongoing studies include oral minoxidil (*n* = 1) and CC (*n* = 1). In addition to new synthetic entities, interesting perspectives are cell therapy and in vitro neogenesis of hair follicles from autologous cells [75].

### 3.3. Palliative Care and Supplementary Management

Beyond the medical approach stricto sensu, it is important to remember that supportive care in oncology is essential through an early and multidisciplinary approach [25,59]. The use of wigs and turbans represents one of the current methods of dealing with established alopecia. Several types exist, using different hair-making technologies. Having an early haircut should be advised for limiting the putative change in self-perception induced by alopecia. For psychological concerns, referral to patient associations and psychological support should also be proposed, depending on the situation [4,5]. As the aesthetic aspect can play a major role in improving QoL, patients should be referred to nurses and socio-aestheticians specialized in these types of cares. The latter can propose different camouflage techniques, such as wigs/turbans, dermopigmentation and keratin powder. Dermopigmentation consists of tattooing micro-dots/-traits on the skin through a dermal injection of bioresorbable pigments, giving the illusion of an increased hair and eyebrow density. The treatment requires an average of 2–3 sessions for an optimal result, lasting 2–5 years. Keratin powder is based on natural keratin attaching to the remaining hair via static electricity, allowing a dramatic but temporary gain in hair density [20,25,59].

## 4. Conclusions

Alopecia, apart from being a well-known side effect of cancer treatments, can reveal a diverse panel of etiologies linked to the disease itself. As such, (bio)clinical investigation should be performed when facing (a risk of) alopecia in the context of cancer in order to offer proper management. ANIA management frequently involves pluridisciplinary management, with the need to assess psychological repercussions and the consequences on the quality of life of patients. Each class of cancer treatment exhibits its own characteristics, and ANIA should not be seen as a homogeneous concern. On the preventive side, eCC has been positioned as a potent tool. Although current treatments exhibit mild efficacy, several therapeutic approaches are under development. 

## Figures and Tables

**Figure 1 cancers-13-05556-f001:**
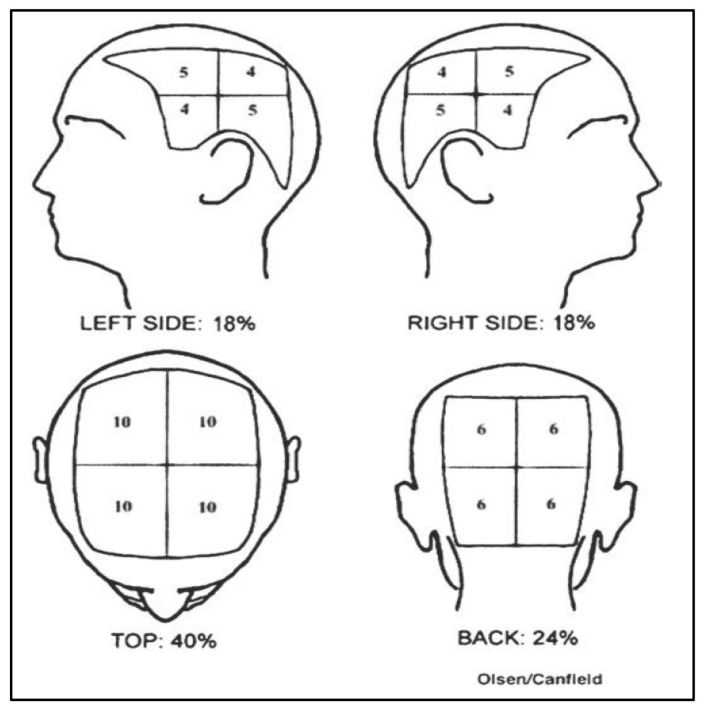
Severity of Alopecia Tool (SALT) score scheme (adapted from reference [16]).

**Table 1 cancers-13-05556-t001:** Main characteristics of hair disorders according to the anticancer therapies used (adapted from references [7,8,9,10,11,12,13,14]).

Treatment Type	Clinical Topography	Main Incriminated Mechanism(s)	Time to Onset	Reversibility	Frequency (%) and Range ([])
Chemotherapies	Diffuse and +/− total	Cell division blockage and apoptosis Destruction of the follicle	2–3 weeks from start	Average: 3–6 months post-treatment Irreversible (with certain regimens)	≈65 [<10–100]
Endocrine therapies	Hair thinning AGA-like pattern	Miniaturization of the follicle	1–91 months	Not systematic	≈5 [0–25]
Targeted therapies	Very variable (target dependent)	Miniaturization of the follicle (+/− destruction)	Very variable	Possible even during treatment Irreversible with some molecules	≈15 [2–60]
Radiotherapy (<43 Gy)	Depending on the radiation field	Destruction of the follicle	1–3 weeks from start	Average: 2–4 months post-irradiation	≈75–100
Radiotherapy (≥43 Gy)	Depending on the radiation field	Destruction of the follicle	≈100 weeks	No (scaring alopecia)	≈75–100
Immunotherapies	Variable	Cycle blockage and dysimmunity	Variable	Variable	≈1–2

**Table 2 cancers-13-05556-t002:** Comparison of SALT, Olsen and CTCAE v5.0 scores to assess the severity of chemotherapy-induced alopecia (adapted from references [16,17]).

Clinical Aspect	Absent	Slight	Moderate		Severe		Total
SALT hair loss (%)	0	1–24	25–49	50–74	75–95	96–99	100
Olsen grades	0	1	2	3	4a	4b	5
CTCAE v5.0 grades	0	1		2			

**Table 3 cancers-13-05556-t003:** Main chemotherapy molecules used against solid tumors and their corresponding frequency of alopecia (adapted from references [10,11]).

Molecule (Class)	All-Grade Estimated Frequency (%)
Daunorubicin, doxorubicin, epirubicin (TI2)	≈80–100
Docetaxel, paclitaxel (taxanes)	
High-dose cyclophosphamide (AA)	
Etoposide, Idarubicin (TI2)	≈40–60
Intravenous topotecan, irinotecan (TI1)	
Bleomycin (CA)	≈10–30
Vinblastine, vincristine, vinorelbine (PA)	
5-Fluorouracil, gemcitabine, methotrexate (AM)	
Capecitabine (AM)	<10
Carboplatin, cisplatin, oxaliplatin (PS)	

Nota bene: frequencies are given for monotherapies and may be susceptible to variation depending on the chemotherapy protocol (with increased frequencies with polychemotherapy protocols). Abbreviations: AA = alkylating agent; AM = antimetabolite; CA = cytostatic antibiotic; PA = periwinkle alkaloids; PS = platinum salts; TI1 = type 1 topoisomerase inhibitor; TI2 = type 2 topoisomerase inhibitor.

**Table 4 cancers-13-05556-t004:** Main * targeted therapy classes used against solid tumors and their corresponding risk of alopecia (adapted from references [10,12,13,14]).

Molecule (Class)	All-Grade Estimated Frequency (%)
SMOi (vismodegib specifically)	60
Mul-I (e.g., sorafenib, regorafenib)	25–30
BRAFi (e.g., dabrafenib, vemurafenib)	20–25
EGFRi (e.g., afatinib, erlotinib)	5–15
VEGFRi (e.g., axitinib, cabozantinib, pazopanib, sunitinib)	
Anti-VEGF (bevacizumab)	
Anti-EGFR (e.g., cetuximab)	
ALKi (e.g., crizotinib)	
MEKi (e.g., trametinib)	

* Nota bene: targeted therapies consist of a constantly increasing panel of molecules; thus, only listed here are the classes (with molecules as examples) that have been reported to exhibit alopecia with a minimal frequency of 5%. Abbreviations: ALK = anaplastic lymphoma kinase; anti- = monoclonal antibody; BRAF = B-rapidly accelerated fibrosarcoma; EGFR = epidermal growth factor receptor; -i = -inhibitor; HER2 = human EGFR 2; MEK = MAPK/ERK (extracellular signal-regulated kinase) kinase; Mul-i = multi-target protein kinase inhibitor; SMO = smoothened; (V)EGF(R) = vascular endothelial growth factor (receptor).

**Table 5 cancers-13-05556-t005:** Main endocrine therapies used against solid tumors and their corresponding frequency of alopecia (adapted from references [8,9,10]).

Molecule (Class)	All-Grade Estimated Frequency (%)
All types of endocrine therapies	≃5%
All types of endocrine therapies *	≃10%
Letrozole (AI) + ribociclib (CDK4/6i)	≃33%
Anastrozole (AI) + gosereline (aGnRH)	≃25%
Letrozole (AI) + palbociclib (CDK4/6i)	≃22%
Fulvestrant (AE) + palbociclib (CDK4/6i)	≃15%
Tamoxifen (SERM) then anastrozole (AI)	≃15%
Tamoxifen (SERM)	≃10%
Leuproreline (aGnRH)	≃10%
Exemestane (AI) + aminoglutethimide	≃10%
AI + fulvestrant (AE)	≃8%
Fulvestrant (AE)	≃2%
Anastrozole, letrozole, exemestane (AI)	≃2%
Flutamide, bicalutamide, nilutamide, abiraterone, enzalutamide (ADT)	≤1%

Nota bene: frequencies for the most commonly used hormone therapies in clinical practice are given. * = when used in combination. Abbreviations: ADT = androgen deprivation therapy; AE = anti-estrogen; GnRHa = gonadotrophin-releasing hormone agonist; AI = aromatase inhibitor; CDK4/6i = cyclin-dependent kinases 4/6 inhibitor; SERM = selective estrogen receptor modulator.

**Table 6 cancers-13-05556-t006:** Currently issued recommendations for the management of ANIA based on the literature (adapted from references [10,11,12,13,59]).

Type of Alopecia	Recommendation(s)	Level of Evidence
Global approaches	Hair status evaluation and differential diagnoses eviction (anamnesis, clinical examination, biological assessment, trichoscopy, trichogram, +/− biopsy) Haircut before treatment initiation Hair prosthesis and textile accessories Camouflage techniques (e.g., pigmentation, keratin powder) Early accompaniment (medical, paramedical, psychologist, cancer support group)	
CIA	Prevention of hair loss: eCC * Acceleration of spontaneous regrowth: TMX 2–5% **	II IV
pCIA	TMX-5% ** (twice daily) Spironolactone ***	IV IV
EIA	TMX-5% ** (twice daily) Spironolactone ***	III IV
RIA	Acceleration of spontaneous regrowth: TMX 2–5% **	IV
pRIA	TMX-5% ** (twice daily) Surgery (e.g., hair graft, skin expansion)	IV IV
IIA	Class IV DC ****	IV
TIA	TMX-5% ** (twice daily) If inflammation is associated: DC	IV IV
Eyebrows/Lashes	Bimatoprost	III

Nota bene: some of these proposed treatments are under investigation and are not yet FDA approved; more detailed information is in the text. Abbreviations: (p)CIA = (persistent) chemotherapy-induced alopecia; DC = dermocorticoids; eCC = electronic cooling cap; EIA = endocrine therapy-induced alopecia; IIA = immunotherapy-induced alopecia; (p)RIA = (persistent) radiotherapy-induced alopecia; TIA = targeted therapy-induced alopecia; TMX = topical minoxidil. Addenda: * Paxman© and Dignicap© scalp cooling systems are FDA approved; ** for women, only the foam type has FDA approval for AGA treatment, and for CIA/RIA, TMX should be used twice daily, during treatment protocol and up to 4 months after; *** in the context of a hormone-sensitive tumor, it requires particular supervision considering the theoretical risk of hormone stimulation; **** treatment proposed for primary IIA.

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
