# Peer review of "Cancer-Related Alopecia: From Etiologies to Global Management"

_cancers, 2021, doi:10.3390/cancers13215556_

Round 1

Reviewer 1 Report

The manuscript presents a review of literature on cancer-related alopecia and the discussed issue is undoubtedly vital. 

I have some major and minor remarks:

  1. In the text there is lack of introduction to the analyzed subject but the authors start the manuscript with well-known general description about the hair physiology and pathology. The authors should rearrange this part and discuss it in the context of cancer-related alopecia.
  2. At the end of the first part (introduction) the authors should include the aim of the study.
  3. I my opinion the III part of the manuscript is out of the scope of this article and could be removed. The described studies in this part mainly concentrate on breast cancer and not other malignancies. 
  4. The part about management is to general. The authors should add more information about each method of treatment (i.e. when to start the treatment, for how long, what doses?)
  5. In the page 6 the sentence "pCIA is defined by the presence of alopecia beyond 6 after ..." is not complete.

Author Response

The manuscript presents a review of literature on cancer-related alopecia and the discussed issue is undoubtedly vital. 

I have some major and minor remarks:

  1. In the text there is lack of introduction to the analyzed subject but the authors start the manuscript with well-known general description about the hair physiology and pathology. The authors should rearrange this part and discuss it in the context of cancer-related alopecia.

We thank the reviewer for this very relevant remark. A such, we modified the introduction in order to offer a clearer scope of the review, shortened the physiological side and added a graphical abstract. Actually the distinct mechanisms according to treatments are discussed in the second section.

  1. At the end of the first part (introduction) the authors should include the aim of the study.

We thank the reviewer for the interesting remark. We added a specific part at the end of the introduction, assessing the aim of the review.

  1. I my opinion the III part of the manuscript is out of the scope of this article and could be removed. The described studies in this part mainly concentrate on breast cancer and not other malignancies. 

We thank the reviewer for this interesting remark and we deleted this part. Although essential for the patient, the part III would indeed need a specific review. As such we removed it.

  1. The part about management is to general. The authors should add more information about each method of treatment (i.e. when to start the treatment, for how long, what doses?)

We thank the reviewer for the interesting remark. The precise description of validated treatments with their protocols is actually furnished in the table 6 for an easier access. Furthermore, we decided to have a broader scope in order to fit for a “physician-friendly” purpose

  1. In the page 6 the sentence "pCIA is defined by the presence of alopecia beyond 6 after ..." is not complete.

We thank the reviewer for this interesting remark and we modified the sentence (the word “month” lacked).

Reviewer 2 Report

The manuscript cancers-1391781, Cancer-related alopecia: from etiologies to global management, is a review of literature targeting a relatively interesting theme.

The manuscript suffers from multiple editing problems that renders its lecture difficult. The authors should use the journal’s template and correct all the editing problems.

The article lacks a proper introduction. The authors should present the scope of the paper. What is its objective? Are there other similar review work in the literature and if so highlight what this work add new. The authors should argue why this review is important.

Multiple statements are given without appropriate references. The authors need to support their statements by proper references.

The authors often simply reiterate the literature without any personal input. A good review should provide insights beyond a summary. I would also recommend the authors to have a more critical view on their data. The review should not be just a collection of data. It should present also contradicting data and new hypotheses.

Author Response

The manuscript cancers-1391781, Cancer-related alopecia: from etiologies to global management, is a review of literature targeting a relatively interesting theme.

The manuscript suffers from multiple editing problems that renders its lecture difficult. The authors should use the journal’s template and correct all the editing problems.

We thank the reviewer for this remark. Actually, we used the Cancers template. Meanwhile, we tried our best to obtain an easier-to-read approach by removing some subsections and with more space.

The article lacks a proper introduction. The authors should present the scope of the paper. What is its objective? Are there other similar review work in the literature and if so highlight what this work add new. The authors should argue why this review is important.

We thank the reviewer for the interesting remark. We actually changed the introduction of the review, with a clearer scope in the introduction and added a graphical abstract. Furthermore we added a specific scope at the end of the introduction.

Multiple statements are given without appropriate references. The authors need to support their statements by proper references.

We thank the reviewer for the interesting remark. Meanwhile, we checked again all of the statements given in the text and they are referenced.

The authors often simply reiterate the literature without any personal input. A good review should provide insights beyond a summary. I would also recommend the authors to have a more critical view on their data. The review should not be just a collection of data. It should present also contradicting data and new hypotheses.

We thank the reviewer for the interesting remark. Actually, the scope of this review is to give a “state-of-the-art” and global approach for clinicians, from molecular insights to comprehensive description of the different cancer-related alopecia with their associated frequencies. To our knowledge, although several reviews exhibited  known validated treatments with their protocols here  is not a proper review giving this physician-guided approach

Round 2

Reviewer 1 Report

The authors have improved the manuscript. In my opinion it may be published in current form

Reviewer 2 Report

The authors changed their manuscript  and improve its quality. There are still major editing problems. The authors should look at any article published in the Cancers journal. For example, the style of table, the caption before table and not after, and so on.